# Alcohol-Supported Cu-Mediated ^18^F-Fluorination of Iodonium Salts under “Minimalist” Conditions

**DOI:** 10.3390/molecules24173197

**Published:** 2019-09-03

**Authors:** Victoriya V. Orlovskaya, Daniel J. Modemann, Olga F. Kuznetsova, Olga S. Fedorova, Elizaveta A. Urusova, Niklas Kolks, Bernd Neumaier, Raisa N. Krasikova, Boris D. Zlatopolskiy

**Affiliations:** 1N.P.Bechtereva Institute of the Human Brain, 197376 St.-Petersburg, Russia (V.V.O.) (O.F.K.) (O.S.F.); 2Institute of Neuroscience and Medicine, INM-5: Nuclear Chemistry, Forschungszentrum Jülich GmbH, 52425 Jülich, Germany (D.J.M.) (E.A.U.) (N.K.) (B.D.Z.); 3Institute of Radiochemistry and Experimental Molecular Imaging, University Clinic Cologne, 50937 Cologne, Germany; 4Max Planck Institute for Metabolism Research, 50931 Cologne, Germany; 5St.-Petersburg State University, 199034 St.-Petersburg, Russia

**Keywords:** ^18^F, radiolabeling, iodonium salts, copper-mediated, positron emission tomography

## Abstract

In the era of personalized precision medicine, positron emission tomography (PET) and related hybrid methods like PET/CT and PET/MRI gain recognition as indispensable tools of clinical diagnostics. A broader implementation of these imaging modalities in clinical routine is closely dependent on the increased availability of established and emerging PET-tracers, which in turn could be accessible by the development of simple, reliable, and efficient radiolabeling procedures. A further requirement is a *c*GMP production of imaging probes in automated synthesis modules. Herein, a novel protocol for the efficient preparation of ^18^F-labeled aromatics via Cu-mediated radiofluorination of (aryl)(mesityl)iodonium salts without the need of evaporation steps is described. Labeled aromatics were prepared in high radiochemical yields simply by heating of iodonium [^18^F]fluorides with the Cu-mediator in methanolic DMF. The iodonium [^18^F]fluorides were prepared by direct elution of ^18^F^−^ from an anion exchange resin with solutions of the corresponding precursors in MeOH/DMF. The practicality of the novel method was confirmed by the racemization-free production of radiolabeled fluorophenylalanines, including hitherto unknown 3-[^18^F]FPhe, in 22–69% isolated radiochemical yields as well as its direct implementation into a remote-controlled synthesis unit.

## 1. Introduction

Widespread implementation of molecular imaging techniques, especially positron emission tomography (PET) and related hybrid methods like PET/CT and PET/MRI in clinical practice have significantly contributed to a considerable increase of diagnostic accuracy in recent years. PET offers the unique opportunity to visualize physiological and pathological processes on the molecular level. This imaging modality utilizes probes labeled with positron emitting radionuclides (PET-tracers) interacting specifically with molecular targets or biochemical processes of interest. Biodistribution of such probes can be determined by the detection of antiparallel γ-photons originating from electron–positron annihilation. PET probes are used for accurate diagnosis and staging of diseases as well as monitoring of therapy success (e.g., for tumors, neurological or cardiac disorders). In addition to clinical applications, PET is a powerful tool in drug development which provides fast and precise assessment of pharmacological properties of drug candidates *in vivo*. The main challenge of PET-chemistry is the short half-life of the majority of commonly used PET radionuclides like ^11^C (20.4 min), ^13^N (10 min) and ^15^O (2 min) which severely limits the scope of reactions which can be used for radiolabeling.

^18^F is the most widely used PET radionuclide owing to the accessibility of no-carrier-added (n.c.a.) ^18^F^–^ in multi-Curie amounts at low and medium energy cyclotrons from [^18^O]H_2_O via the high-yielding ^18^O(p,n)^18^F nuclear reaction. ^18^F has a longer half-life (109.8 min) which enables extended PET acquisition protocols and shipping of radiofluorinated tracers to remote PET centers as well as favorable decay characteristics like a high positron branching ratio of 97% and a low positron energy (E_βmax_ 0.63 MeV). The latter enables imaging with high spatial resolution.

The vast majority of ^18^F-labeled PET radiotracers are produced by S_N_2 and S_N_Ar radiofluorination reactions. In order to be applied for these reactions ^18^F^–^ produced from [^18^O]H_2_O has to be transformed into anhydrous [^18^F]F^-^ with high nucleophilicity. Conventionally, in order to separate the bulk of [^18^O]water, ^18^F^–^ is trapped on an anion exchange resin. It is recovered using an aqueous solution of suitable bases like K_2_CO_3_, Cs_2_CO_3_ or tetraalkylammonium hydrogen carbonates. [^18^O]H_2_O is then removed by repeatedly time-consuming azeotropic drying with MeCN. Finally, the residual anhydrous [^18^F]fluoride salt is taken up in a solution of the radiolabeling precursor in an aprotic polar solvent and heated for a short time affording the desired radiolabeled product. If potassium salts are used, K^+^ chelators like K_2.2.2_ or 18-crown-6 which increase the solubility of ^18^F^–^ in organic solvents and its nucleophilicity by virtue of the charge separation, are typically added. Whereas a broad spectrum of labeled aliphatic probes can be prepared, only highly electron deficient (hetero)aromatics containing electron-withdrawing groups in *o*- or *p*-position to the leaving group can be labeled using such approach. Recently developed protocols, e.g., for Pd- [1,2,3], Ni- [4,5], and especially Cu-mediated radiofluorination enabled to overcome this limitation and efficiently prepare electron-neutral and even -rich ^18^F-labeled (hetero)aromatics. Initially described for (aryl)(mesityl)iodonium salts (Scheme 1) [6], Cu-mediated radiofluorination was also applied for labeling of easily accessible aryl pinacol boronates (ArBPin) [7,8], aryl boronic acids [9], aryl trialkyl stannanes [10] as well as (aryl)(mesityl)iodonium salts generated in situ from electron-rich (hetero)aromatics [11] and directed CH-^18^F-fluorination [12]. The original procedures, while working well at a small scale, were in some cases inoperative for batch PET-tracer production. Several radiolabeling protocols, which allowed overcoming this limitation, were published [13,14,15,16,17,18,19]. Moreover, the “minimalist protocol” originally developed for labeling of suitable onium salt precursors using only [^18^F]fluoride [20] was successfully implemented into Cu-mediated radiofluorination of (aryl)(mesityl)iodonium salts. Additionally, this protocol was also transferred to an automated synthesis [17,21]. This enabled the high-yielding preparation of several clinically relevant PET-tracers like, 4-[^18^F]fluorophenylalanine, 6-[^18^F]fluorodopamine and [^18^F]DAA1106 on a preparative scale using only Cu-catalyst, iodonium salt and [^18^F]fluoride. The method obviates the need for azeotropic drying, base and any other additives. Accordingly, [^18^F]fluoride is directly eluted from a anion exchange resin with a solution of the corresponding iodonium salt precursor in MeOH. Low-boiling MeOH is evaporated; the residue is taken up in a solution of Cu(MeCN)_4_OTf in DMF and heated furnishing the desired radiolabeled product. However, solvent evaporation before ^18^F-fluorination still has to be carried out. Noteworthy, in the case of remote-controlled radiosyntheses, complete removal of MeOH is a prerequisite to obtain high radiolabeling yields. In contrast, the “alcohol-enhanced” Cu-mediated radiofluorination not only enables to obtain considerably higher radiochemical yields (RCYs) but also substantially simplifies the production of ^18^F-fluorinated (hetero)aromatics by the elimination of any evaporation steps. Furthermore, a broad scope of stannyl and boronyl substrates could be efficiently radiolabeled under general reaction conditions [22,23]. Hence, ^18^F^–^ is directly eluted with an alcoholic (usually *n*BuOH) solution of a suitable salt like Et_4_NHCO_3_, Bu_4_POMs, K_2_CO_3_/K2.2.2, etc., into a solution of the appropriate precursor and Cu(py)_4_(OTf)_2_ in DMA or DMF. The resulting solution is briefly heated affording the ^18^F-labeled probe. The efficacy of Cu-mediated ^18^F-fluorination in alcoholic media markedly contradicted previous observations concerning the deleterious effect of protic solvents, including alcohols, on S_n_Ar fluorination. Noteworthy, quite recently the efficient metal-free ^18^F-fluorination of electron-deficient (hetero)aromatics in pure EtOH was published [24]. Herein, we disclose a novel evaporation step free protocol for Cu-mediated radiofluorination of (aryl)(mesityl)iodonium salts which combines advantages of the “alcohol-enhanced” with the “minimalist” approach. We applied the procedure for radiolabeling of model substrates and also for the preparation of [^18^F]fluorophenylalanines ([^18^F]FPhe) including hitherto unknown 3-[^18^F]FPhe.

## 2. Results and Discussion

(Mesityl)[(4-methoxy)phenyl]iodonium tosylate and tetrafluoroborate (**1**·OTs and **1**·BF_4_) were selected as model substrates (Scheme 2). First, the elution of [^18^F]fluoride from an anion exchange resin with **1**·OTs or **1**·BF_4_ in anhydrous or 80% aqueous DMF as well as in a mixture of DMF with different alcohols (20% alcohol content) was examined (Figure 1). Experiments were carried out either manually or in a self-made semi-automated synthesis module. The highest recovery of ^18^F^–^, was obtained with aqueous and methanolic DMF, amounting to 83% and 73%, respectively (with **1**·BF_4_). In contrast to the recovery of [^18^F]fluoride with solutions of ammonium and phosphonium salts in ROH/DMF,[22] ^18^F^–^ recovery with solutions of iodonium salts in mixtures of DMF and higher alcohols was only moderate and amounted to 50% and 31% for EtOH and *n*PrOH, respectively (with **1**·BF_4_). Elution efficacy with **1**·BF_4_ in 20% *n*BuOH in DMF was comparably low as with a solution of this precursor in pure DMF (about 20%). Notably, ^18^F^–^ recovery with **1**·OTs in MeOH or EtOH was up to 20% higher than recovery with **1**·BF_4_ in the same solvents. The eluates were directly added to solutions of Cu(MeCN)_4_OTf in DMF (10% final concentration of ROH) and heated at 85 °C for 20 min.

Whereas almost quantitative ^18^F-incorporation yields (>94%) were observed for both radiolabeling substrates using DMF and ROH/DMF mixtures as reaction medium, no radiofluorination took place in aqueous DMF, presumably, owing to the decomposition of Cu(MeCN)_4_OTf in the presence of water and/or unfavorably strong solvation of ^18^F^–^. In trying to further improve ^18^F^–^ recovery, the MeOH content of the elution solution was increased to 30% (12% final concentration of MeOH in the reaction mixture; Figure 2). This modification led to substantially improved elution efficacy (90%), while radiochemical incorporation (RCC) remained high (86%). A further increase of the MeOH content was detrimental and caused a rapid decrease of the ^18^F-incorporation rate owing to the formation of sTable 18F^–^(MeOH)_n_ clusters (n = 2–8, mainly 3, 4). Cluster formation substantially decreases the nucleophilicity of fluoride [25].

Next, the influence of aprotic solvents was studied. Replacement of DMF by CH_3_CN, DMSO, DMA, DMI, DMPU or sulfolane had no significant influence on ^18^F^-^ recovery (73–90%). However, surprisingly, no formation of 4-[^18^F]fluoroanisole ([^18^F]**2**) was observed in any of the examined solvents including DMA, a higher analog of DMF which was the best suited solvent for Cu-mediated radiofluorination of boronate and stannyl substrates (data not shown). This somewhat unexpected observation could be possibly explained by the fact that DMF, in contrast to all other studied solvents, is a monodentate *O*-donor ligand which can stabilize the intermediately formed (Ar)(MesI)Cu(III)^18^F complex and facilitate its reductive elimination [26].

Interestingly, in contrast to earlier observations,[17] aerobic conditions were not a prerequisite for successful radiolabeling. The same RCCs were obtained under air and N_2_ indicating a direct oxidation of the Cu(I) complex by the iodonium salt precursor (data not shown).

Additionally, radiolabeling kinetics was briefly studied. Incorporation yields of >90 % were already observed after 5 min incubation at 85 °C (Figure 3).

Furthermore, the dependency of ^18^F^–^ elution efficiency and RCC of [^18^F]**2** at 85 °C after 20 min reaction time on the amount of **1**·OTs precursor was studied (Figure 4). In these experiments equimolar amounts of Cu(MeCN)_4_OTf were used. While ≥90% ^18^F^-^ recovery was observed for the different precursor amounts (3.5–21 µmol), maximum RCCs >90% were obtained with 21 and 14 µmol of **1**·OTs. Minimization of the amount of iodonium salt from 7 to 3.5 µmol still produced [^18^F]**2** in RCCs of 81% and 44 %, respectively.

The scope of the novel radiofluorination protocol was further evaluated using phenyl- and (3-carbonyl)phenyl-substituted (mesityl)iodonium tetrafluoroborates as a model for an electron-neutral and an electron-deficient substrate, respectively. Gratifyingly, both precursors were successfully radiolabeled affording [^18^F]fluorobenzene ([^18^F]3) and 3-[^18^F]fluorobenzaldehyde ([^18^F]4) in RCCs of >90% and >72%, respectively (Table 1). Accordingly, using the novel protocols, model compounds were prepared in RCCs comparable to or even better than those obtained applying the original “minimalist” protocol for Cu-mediated radiofluorination [17].

Being particularly interested in fast and efficient procedures for the preparation of ^18^F-labeled aromatic amino acids, we applied the novel protocol for the synthesis of 2–4-[^18^F]FPhes ([^18^F]**8**–**10**) (Table 1, Scheme 3), which should be potentially suitable for imaging of tumors and neurological disorders.

The precursor of 3-[^18^F]FPhe **12** was prepared as follows (Scheme 4). (*S*)-Ni-BPB-Gly [27] was alkylated with 3-iodobenzyl bromide, affording the Ni(II) complex **14** as a single (*S*,*S*)-isomer. **14** was decomposed by transchelation with the ditetraethylammonium salt of DTPA [28] furnishing 3-IPhe, which was *N*-Boc and *O*-*t*Bu protected in two sequential steps to give Boc-3-IPhe-O*t*Bu **15** in 56% over four steps with a single chromatographic purification. The latter was transformed into Boc-3-(SnMe_3_)Phe-O*t*Bu which in turn was allowed to react with MesI(OH)OTs affording the iodonium salt **12** in 20% yield over six steps.

*N*-mono-Boc protected **12** and **13** as well as *N*,*N*-di-Boc protected precursors of 2-[^18^F]FPhe and 4-[^18^F]FPhe, **11** and **14** [28], were successfully radiolabeled affording protected radiolabeled FPhes, in RCCs of 85-86%, 62%, 59–68%, and 80%, respectively (Table 1; Scheme 3). Finally, deprotection with 12 n HCl and HPLC purification furnished 2–4-[^18^F]FPhe in isolated RCYs of 22–30%, 37–43%, and 67–69%, respectively, within 110 min. The molar activity determined for 4-[^18^F]FPhe (2.22 GBq), which was produced from the iodonium salt **13**, amounted to 207 GBq/µmol. Importantly, the novel procedure enabled the production of 3- and 4-[^18^F]FPhes in a remote-controlled synthesis unit. Remarkably, whereas, the application of the conventional “minimalist” protocol led to significant loss of enantiopurity [formation of up to 20% of the (*R*)-isomer] in the case of 2-[^18^F]FPhe [28], the novel procedure delivered this radiotracer with >95% enantiomeric excess (*ee*). The observed suppression of racemization in the case 2-[^18^F]FPhe could be explained as follows. The thermally unstable iodonium bicarbonate and carbonate iodonium salts could form as a result of a partial anion exchange with the anion exchange resin in HCO_3_^–^ and/or CO_3_^2–^ form during the elution of [^18^F]fluoride. In the subsequent MeOH evaporation step CO_2_ and H_2_O will be lost affording, in the case of the *ortho*-iodonium salt **9**, the corresponding betaine with concurrent loss of stereointegrity (refer to [28] for a more detailed discussion). Consequently, avoiding MeOH evaporation suppresses racemization. Accordingly, the novel radiolabeling method could be beneficial also for the production of other *ortho*-[^18^F]fluoro-substituted aromatic amino acids such as 6-[^18^F]FDOPA, 2-[^18^F]FTyr and 6-[^18^F]FMT with high enantiomeric purity. 

## 3. Materials and Methods

### 3.1. General

Chemicals and solvents were purchased from Sigma-Aldrich GmbH (Steinheim, Germany), Fluka AG (Buchs, Switzerland), TCI EUROPE N.V. (Zwijndrecht, Belgium), ChemPUR GmbH (Karlsruhe, Germany), Merck KGaA (Darmstadt, Germany) and ABCR GmbH (Karlsruhe, Germany) and used as delivered. Anhydrous solvents were purchased from Sigma-Aldrich GmbH (Steinheim, Germany) and stored under argon. 

### 3.2. Nuclear Magnetic Resonance

^1^H-NMR spectra: Bruker Avance III (400 MHz) and Varian INOVA 400 (400 MHz). ^1^H chemical shifts are reported in ppm relative to residual peaks of deuterated solvents. The observed signal multiplicities are characterized as follows: s = singlet, d = doublet, t = triplet, m = multiplet, and br = broad. Coupling constants (J) were reported in Hertz (Hz). ^13^C-NMR spectra [additional APT (Attached Proton Test) or DEPT (Distortionless Enhancement by Polarization Transfer)]: Bruker Avance III (101 MHz) and Varian INOVA 400 (101 MHz). ^13^C chemical shifts are reported in ppm relative to residual peaks of deuterated solvents. ^19^F-NMR spectra: and Bruker DPX Avance 200 (188 MHz). Copies of the ^1^H and ^13^C NMR spectra are available in the Appendix A.

### 3.3. Mass Spectroscopy

High-resolution mass spectra (HRMS) were measured on LTQ FT Ultra (Thermo Fisher Scientific Inc., Bremen, Germany). Copies of MS spectra are available in the Appendix A.

### 3.4. Chemistry

All reactions were carried out with magnetic stirring, if not stated otherwise, and, if air or moisture sensitive, substrates and/or reagents were handled in flame-dried glassware under argon or nitrogen. Organic extracts were dried with anhydrous MgSO4.

Column chromatography: Merck silica gel, grade 60, 230–400 mesh. Solvent proportions are indicated in a volume/volume ratio.

Thin layer chromatography (TLC) was performed using precoated sheets, 0.25 mm Sil G/UV254 from Merck KGaA (Darmstadt, Germany). The chromatograms were viewed under UV light (λ = 254 nm).

(4-Methoxyphenyl)(mesityl)iodonium tosylate (**1**·OTs) [29], (4-methoxyphenyl)(mesityl)iodonium tetrafluoroborate (**1**·BF_4_) [17,29], (phenyl)(mesityl)iodonium tetrafluoroborate [30], (3-carbonylphenyl)(mesityl)iodonium tetrafluoroborate [30], 2-{2-[(2*S*)-2-{bis[(*tert*-butoxy)carbonyl]amino}-3-(*tert*-butoxy)-3-oxopropyl]phenyl}(2,4,6-trimethylphenyl)iodanium tetrafluoroborate (**11**) [28], 4-{2-[(2*S*)-2-{[(*tert*-butoxy)carbonyl]amino}-3-(*tert*-butoxy)-3-oxopropyl]phenyl}(2,4,6-trimethylphenyl)iodanium tetrafluoroborate (**13**) [17], 4-{2-[(2*S*)-2-{bis[(*tert*-butoxy)carbonyl]amino}-3-(*tert*-butoxy)-3-oxopropyl]phenyl}(2,4,6-trimethylphenyl)iodanium tetrafluoroborate [28], (*S*)-Ni-BPB-Gly [27] hydroxy{[(4-methylphenyl)sulfonyl]oxy}(2,4,6-trimethylphenyl)-λ3–iodine [31] were prepared according to the literature or modified literature procedures.

#### 3.4.1. (*S*,*S*)-Ni-BPB-3-IPhe (**14**)

A suspension of (*S*)-Ni-BPB-Gly (5 g, 10.04 mmol) in a mixture of DMF (5 mL) and MeCN (10 mL) was degassed with three freeze-pump-thaw cycles at −70 °C. Afterwards, NaH (520 mg, 60 % in oil, 13.0 mmol) and 3-iodobenzyl bromide (3 g, 10.10 mmol) were sequentially added. The cooling bath was replaced by another one (–10 °C; ice–salt bath), and the reaction mixture was vigorously stirred for 40 min. Aqueous AcOH (5 mL + 8 mL H_2_O) was carefully added, the mixture was poured into ice water (1 L) and the resulting mixture was stirred 16 h. The formed fine crystalline precipitate was filtered off, washed with H_2_O, dried and carefully washed with Et_2_O to give **14** as a red solid [6.4 g, 89%, 5% of (*S*,*R*)-isomer according to the ^1^H-spectrum]. ^1^H NMR (400 MHz, CDCl_3_) δ 1.77–1.89 (m, 1 H) 1.97–2.08 (m, 1 H) 2.32–2.46 (m, 2 H) 2.46–2.63 (m, 1 H) 2.82 (dd, *J* = 13.8, 6.0 Hz, 1 H) 3.03 (dd, *J* = 13.8, 4.4 Hz, 1 H) 3.14–3.24 (m, 1 H) 3.35 (dd, *J* = 9.8, 7.2 Hz, 1 H) 3.45–3.52 (m, 1 H) 4.23 (dd, *J* = 6.0, 4.4 Hz, 1 H) 4.30 (d, *J* = 12.6 Hz, 1 H) 6.68 (d, *J* = 3.9 Hz, 2 H) 6.86 (d, *J* = 7.6 Hz, 1 H) 7.10–7.13 (m, 2 H) 7.13–7.20 (m, 2 H) 7.28–7.35 (m, 3 H) 7.42–7.48 (m, 2 H) 7.54–7.58 (m, 2 H) 7.72 (ddd, *J* = 5.3, 3.6, 1.8 Hz, 1 H) 8.02 (d, *J* = 7.1 Hz, 2 H) 8.26 (d, *J* = 8.6 Hz, 1 H). ^13^C NMR (101 MHz, CDCl_3_) δ 180.3, 178.2, 171.3, 142.9, 139.1, 138.3, 136.5, 134.0, 133.5, 133.2 (×2), 132.51, 131.47, 130.3, 129.9, 129.7, 129.2, 129.0, 128.9, 128.80, 128.77, 127.7, 127.1, 125.92, 123.4, 120.6, 95.1, 71.1, 70.3, 63.3, 57.3, 39.4, 30.8, 23.4. HRMS (ESI): *m*/*z* [M + H]^+^ calcd for C_34_H_31_O_3_N_3_INi^+^: 714.07584; found: 714.07572, [M + Na]^+^ calcd for C_34_H_31_O_3_N_3_INiNa^+^: 736.05778; found: 736.05790. Correct isotopic pattern.

#### 3.4.2. Boc-3IPhe-OtBu (**15**)

A solution of DTPA(Et_4_N)_2–3_ [108 mL; prepared by dissolution of DTPA (28.1 g, 71.43 mmol) and Et_4_NHCO_3_ (38.2 g, 199.7 mmol) in 100 mL H_2_O] was added to a solution of **13** (6.3 g, 8.84 mmol) in MeOH (200 mL) and the resulting dark red mixture was stirred at 70 °C for 24 h. At this point, the TLC control (CHCl3:acetone = 3:1, visualization: UV and ninhydrin) showed the almost complete decomposition of the Ni complex **13** (color change: dark red to blue). MeOH was distilled off under reduced pressure, the pH of the resulting suspension was adjusted to 9 with 1 M NaOH, the precipitate was filtered, washed with an ice-cold H_2_O and dried affording (*S*)-2-[*N*-(*N*’-benzyl-prolyl)amino]benzophenone (BPB). The filtrate was extracted with Et_2_O (×3; if the precipitation of the Et_2_O insoluble 3-IPhe was observed, more 1 m NaOH was added until the amino acid was completely dissolved). The organic extracts were washed with brine (×2), dried and concentrated under reduced pressure to give the second crop of BPB. 10% NaHCO_3_ (30 mL) was added to the aqueous fraction followed by Boc_2_O (5.7 g, 26.15 mmol). Afterwards MeOH was added until a homogenous solution was obtained and the resulting mixture was stirred for 16 h. MeOH was removed under reduced pressure, the residual aqueous solution was extracted with Et_2_O (×3), the organic fractions were discarded and the aqueous fraction was carefully acidified to pH 2 with solid NaHSO_4_ (vigorous evolution of CO_2_). The formed emulsion was extracted with Et_2_O (×2), the organic fraction washed with 1 m NaHSO_4_ (×3), H_2_O (×3), brine (×2), dried and concentrated under reduced pressure affording crude Boc-3IPhe-OH (2.5 g, 72%) as a colorless oil gradually solidified into a colorless solid which was used in the next step without further purification. *R*_f_: 0.3 [EtOAc:hexane = 1:1.5 (3% AcOH)]. ^1^H NMR (400 MHz, DMSO-*d*_6_, mixture of two rotamers; only spectrum of the major rotamer is provided) δ ppm 1.32 (s, 9 H) 2.77 (dd, *J* = 13.6, 10.8 Hz, 1 H) 2.99 (dd, *J* = 13.8, 4.3 Hz, 1 H) 4.08 (dd, *J* = 19.0, 4.4 Hz, 1 H) 4.08 (d, *J* = 2.4 Hz, 1 H) 7.02–7.18 (m, 2 H) 7.27 (d, *J* = 7.8 Hz, 1 H) 7.57 (d, *J* = 7.8 Hz, 1 H) 7.62 (s, 1 H) 12.24–13.03 (br, 1 H).

*tert*-Butyl 2,2,2-trichloroacetimidate (2.77 mL, 3.38 g, 15.49 mmol) was added to a suspension of Boc-3IPhe-OH (2.4 g, 6.13 mmol) in anhydrous CH_2_Cl_2_ (10 mL) and the reaction mixture was stirred at 40–45 °C for 72 h. The mixture was cooled to ambient temperature, diluted with pentane (40 mL) cooled to −20 °C and filtered. The filtrate was washed with 5% NaHCO_3_ (×3), 1 m NaHSO_4_ (×3), H_2_O (×3), brine (×2), dried and concentrated under reduced pressure. The residual oil was dried at 60 °C and 1 mbar to remove a bulk of the unreacted *tert*-butyl 2,2,2-trichloroacetimidate and the residue was purified by column chromatography (EtOAc: hexane = 1:5) and the product-containing fractions were concentrated under reduced pressure to give colorless oil which was dissolved in pentane. The resulting solution was cooled to 5 °C, filtered and concentrated affording **15** as colorless oil which gradually solidified into a colorless solid. Finally, recrystallization from hexane (the mother liquor was concentrated under reduced pressure and the residue recrystallized from hexane; ×2) furnished the pure **15** [32] (2.37 g, 56% over 4 steps). *R*_f_: 0.5 (EtOAc:hexane = 1:6). ^1^H NMR (400 MHz, CDCl_3_; mixture of two rotamers; only spectrum of the major rotamer is provided) δ 1.42 (s, 9 H) 1.44 (s, 8 H) 3.01 (d, *J* = 5.50 Hz, 2 H) 4.33–4.54 (m, 1 H) 5.04 (d, *J* = 6.75 Hz, 1 H) 6.97–7.09 (m, 1 H) 7.16 (d, *J* = 7.63 Hz, 1 H) 7.53 (s, 1 H) 7.57 (d, *J* = 7.88 Hz, 1 H). ^13^C NMR (101 MHz, CDCl_3_) δ 170.5, 155.0, 138.9, 138.5, 135.8, 130.0, 128.8, 94.2, 82.4, 79.8, 54.7, 38.0, 28.3, 29.0. HRMS (ESI): *m*/*z* [M + H]^+^ calcd for C_18_H_27_INO_4_^+^: 448.09793; found: 448.09800, [M + Na]^+^ calcd for C_18_H_26_INO_4_Na^+^: 470.07987; found: 470.07993.

#### 3.4.3. 3-{2-[(2*S*)-2-{[(*tert*-Butoxy)carbonyl]amino}-3-(*tert*-butoxy)-3-oxopropyl]phenyl}(2,4,6-trime-thylphenyl)iodanium tetrafluoroborate (**12**)

Sn_2_Me_6_ (1.27 mL, 2.01 g, 6.15 mmol) was added to a solution of Boc-3-IPhe-O*t*Bu (**14**) (1.1 g, 2.46 mmol) and Pd(PPh_3_)_4_ (0.284 g, 0.25 mmol) in anhydrous 1,4-dioxane (5 mL) in a glove box. The reaction mixture was stirred at 80 °C for 16 h and thereafter at 110 °C for 3 h, cooled to ambient temperature, filtered through Celite^®^ and concentrated under reduced pressure. The residue was purified by column chromatography (EtOAc:hexane = 1:6 on silica gel with 0.1% CaO) to give Boc-3-(SnMe_3_)Phe-O*t*Bu as a colorless oil (1.1 g, 92%) which was used in the next step without further purification and characterisation. *R*_f_ = 0.6, EtOAc:hexane = 1:6. 

Hydroxy{[(4-methylphenyl)sulfonyl]oxy}-(2,4,6-trimethylphenyl)-λ_3_–iodine (0.98 g, 2.25 mmol) was added to a solution of Boc-3-(SnMe_3_)Phe-O*t*Bu (1.09 g, 2.25 mmol) in CH_2_Cl_2_ (10 mL) and the reaction mixture was stirred for 90 min. After that the mixture was concentrated under reduced pressure. The residue was purified by column chromatography (CH_2_Cl_2_:MeOH = 10:1) to give 3-{2-[(2*S*)-2-{[(*tert*-butoxy)carbonyl]amino}-3-(*tert*-butoxy)-3-oxopropyl]phenyl}(2,4,6-trimethylphenyl)iodanium tosylate as a viscous yellow oil which was immediately used in the next step. *R*_f_ = 0.4, CH_2_Cl_2_:MeOH = 10:1. 

To a solution of tosylate salt in CH_2_Cl_2_ (20 mL) the saturated solution of NaBF_4_ (10 mL) was added and the mixture was vigorously stirred for 10 min. The aqueous solution and precipitate were separated off, the saturated NaBF_4_ (10 mL) was added and the mixture was vigorously stirred for 10 min (×5). The organic fraction was dried and concentrated under reduced pressure. The residue was recrystallized from CH_2_Cl_2_/Et_2_O to give after drying under reduced pressure **12** (0.67 g, 36% over two steps) as a faint yellow solid. ^1^H NMR (400 MHz, CD_3_CN) δ 1.32 (s, 9 H) 1.36 (s, 9 H) 2.34 (s, 3 H) 2.62 (s, 6 H) 2.89 (dd, *J* = 14.0, 9.0 Hz, 1 H) 3.10 (dd, *J* = 13.9, 5.3 Hz, 1 H) 4.22 (d, *J* = 5.7 Hz, 1 H) 7.76 (s, 1H) 5.53 (d, *J* = 6.7 Hz, 1 H) 7.22 (s, 2 H) 7.42 (t, *J* = 7.8 Hz, 1 H) 7.51 (d, *J* = 7.5 Hz, 1 H), 7.73 (d, *J* = 8.3 Hz, 1 H) 7.76 (s, 1 H). ^13^C NMR (101 MHz, CD_3_CN) δ 171.4, 156.3, 145.9, 143.9, 143.6, 135.9, 134.8, 133.6, 133.2, 131.4, 121.4, 112.6, 82.5, 80.0, 56.1, 38.0, 28.4, 28.1, 27.3, 21.0. ^19^F NMR (376 MHz, CD_3_CN) δ −151.32 (^10^BF_4_^−^), −151.37 (^11^BF_4_^−^). HRMS (ESI): *m*/*z* [M]^+^ calcd for C_27_H_37_INO_4_^+^: 566.1754; found: 566.1756.

### 3.5. Radiochemistry

#### 3.5.1. General

All radiosyntheses were carried out using anhydrous DMF (Aldrich) and MeOH (Aldrich or Acros). Cu(MeCN)_4_OTf (Aldrich) was stored under argon. QMA cartridges (Sep-Pak Accell Plus QMA Carbonate Plus Light Cartridge) were obtained from Waters (Waters GmbH, Eschborn, Germany) and used without any preconditioning. RP-cartridges (Strata™-X 33 µm polymeric reversed phase, 200 mg/3 mL, tube) were from Phenomenex (Phenomenex Ltd., Aschaffenburg, Germany).

[^18^F]Fluoride was produced by the ^18^O(p,n)^18^F reaction by bombardment of enriched [^18^O]water with 16.5 MeV protons at the BC1710 cyclotron (The Japan Steel Works Ltd., Shinagawa, Japan) of the INM-5 (Forschungszentrum Jülich) or PETtrace 4 cyclotron (GE Healthcare, Uppsala, Sweden) at the IHB (Saint-Petersburg). If not otherwise noted, all radiolabeling experiments were carried out under ambient or synthetic air.

Standard deviations were calculated using standard formulae.

#### 3.5.2. TLC Analysis

TLC analysis was carried out on silica gel plates 60 F_254_ (Merck), TLC Al foils (Sigma Aldrich) or Polygram SIL G/UV_254_ (Macherey Nagel); radioactivity distribution was measured using a MiniGita radioTLC scanner (Raytest). An aliquot of the quenched reaction mixture (2–3 µL) was applied to a TLC plate; the plate was eluted with 4:1 hexane/EtOAc (A) or 9:1 hexane/EtOAc (B). After developing the plate was immediately covered by tape to prevent the losses of the volatile radioactive products. The *R*_f_ values of the different compounds are compiled in Table 2. The RCC (radiochemical conversion) was determined by dividing of the product peak area by the total peak area. TL-chromatograms can be found in the Appendix A.

#### 3.5.3. HPLC Analysis

Before the determination of radiochemical conversions (RCCs), reaction mixtures were diluted with H_2_O (1–4 mL) (or 50% MeOH) to dissolve any ^18^F-fluoride adsorbed onto the reaction vessel walls. The loss of radioactivity on the vessel walls did not exceed 13% ± 2% from the starting activity (n > 100). All radiochemical yields (RCYs) were decay corrected and radiochemical purities (RCPs) were determined after purification.

High-performance liquid chromatography (HPLC) was carried out using a system equipped with: a Knauer pump, a Knauer K-2500 UV/VIS detector (Knauer), a manual Rheodyne injector (20 μL loop) and a NaI(Tl) well-type scintillation detector (EG&G Ortec; model 276 Photomultiplier Base) with an ACE Mate Amplifier and BIAS supply (all from Ortec Ametek). Table 3 contains the different applied HPLC methods (column and eluent).

Data acquisition and interpretation were performed using Gina software (Raytest). The dead times of the HPLC columns were determined using thiourea. UV and radioactivity detectors were connected in series, giving a time delay of 0.05–0.3 min depending on the flow rate. ^18^F-Labeled compounds were identified by co-injection of the unlabeled reference compounds. The completeness of the radioactivity elution was checked by analyzing of the same sample amount choosing a column bypass (Figure 5).

Alternatively, analytical HPLC was performed on a Dionex ICS- 5000 system equipped with a gradient pump, a variable wavelength UV detector Dionex UV 254 and NaI(Tl) well-type scintillation detector (Carroll and Ramsey Associates, model 105-S). The k’ values of the different radiolabeled compounds with the used HPLC methods are compiled in Table 4. HPL chromatograms can be found in the Appendix A.

#### 3.5.4. Determination of Enantiomeric Purity

Enantiomeric excess was determined by HPLC (method D (Table 3)); cf. Appendix A.

#### 3.5.5. Manual Radiosynthesis. Screening of the Reaction Conditions and Reaction Scope—General Procedure 1 (GP 1)

A solution of [^18^F]F^–^ (0.05–0.5 GBq) in [^18^O]H_2_O was loaded from the male side on a QMA cartridge. The cartridge was flushed from the male side with MeOH (1 mL). ^18^F^–^ was eluted from the female side of the cartridge with a solution of the respective radiolabeling precursor (21 µmol; if not otherwise stated) in a solution of 19% MeOH in DMF (620 µL, MeOH/DMF = 120/500 µL, if not otherwise stated) directly into a solution of Cu(MeCN)_4_OTf (7.9 mg, 21 µmol; if not otherwise stated) in DMF (600 µL; if not otherwise stated) and the resulting solution was heated at 85 °C for 20 min (if not otherwise stated). The reaction mixture was diluted with H_2_O or 50% MeOH (1–4 mL) and RCCs were determined by TLC and/or HPLC (method A and method B of Table 3).

#### 3.5.6. Manual Radiosynthesis. Preparation of 2-[^18^F]FPhe ([^18^F]8) and 4-[^18^F]FPhe ([^18^F]10)–General Procedure 2 (GP 2)

A solution of [^18^F]F^–^ (0.5–5 GBq) in [^18^O]H_2_O was loaded from the male side on a QMA cartridge. The cartridge was flushed from the male side with MeOH (1 mL). ^18^F^–^ was eluted from the female side of the cartridge with a solution of the respective radiolabeling precursor (21 µmol) in 19% MeOH in DMF (620 µL, MeOH/DMF = 120/500 µL) directly into a solution of Cu(MeCN)_4_OTf (7.9 mg, 21 µmol) in DMF (600 µL) and the resulting solution was heated at 85 °C for 20 min. The reaction mixture was diluted with 50% MeOH (4 mL) and stirred for 30 sec. The resulting solution was loaded onto a RP cartridge, the reaction vial was rinsed with H_2_O (2 mL) which was passed through the cartridge. The cartridge was washed with H_2_O (4 mL) and radiolabeled intermediates were eluted MeCN (3 mL). All volatiles were removed under reduced pressure of Ar. 12 n HCl (0.8 mL) was added and the reaction mixture was heated for 10 min at 80 °C. Afterwards the mixture was concentrated under reduced pressure to approximately 0.3–0.4 mL and diluted with H_2_O (0.3 mL). The desired radiolabeled amino acids were isolated by HPLC using method B and C (Table 3).

#### 3.5.7. Semi-Automated Synthesis of ^18^F-Labeled Products–General Procedure 3 (GP 3) (Figure 6)

1. Loading of [^18^F]fluoride (0.1–10 GBq) onto a QMA ion exchange cartridge;

2. Washing of the cartridge with MeOH (2.0 mL);

3. Elution of [^18^F]fluoride from the ion exchange cartridge with a solution of the radiolabeling precursor (21 µmol) in 20% MeOH in DMF (0.6 mL) into RV1 with a solution of Cu(MeCN)_4_OTf (21 µmol) in DMF (0.6 mL);

4. Open valve 1, 2, 3 to completely transfer methanolic solution from QMA to RV1;

5. Heating of the reaction mixture in RV1 at 85 ^°^C for 20 min;

6. Cooling of RV1 down to 50 ^°^C;

7. Addition of water (4 mL) → in the case of [^18^F]-amino acids: precipitation of precursor;

For optimization experiments:

8. Analysis of the reaction mixture by TLC and/or HPLC.

For the production of radiolabeled amino acids [^18^F]**8**–**10**:

8. Loading of the mixture onto a *t*C18 long RP SPE cartridge;

9. Rinsing of the cartridge with H_2_O (5 mL);

10. Elution of the radiolabeled intermediate into RV2 using MeOH (2.0 mL);

11. Place vial RV2 in the heating block;

12. Evaporation of MeOH at 100 ^°^C within 5 min using a flow of N_2_;

13. Addition of 12 n HCl (0.5 mL) and heating at 130 ^°^C for 10 min;

14. Cooling of RV2 to 55 ^°^C and addition of a 10 mM AcONa/50 mM AcOH/0,1 g/L ascorbic acid solution (3.5 mL);

15. Loading of the mixture in the loop of semi preparative HPLC system; HPLC: pump: SYCAM S1122 solvent delivery system, UV detector and radioactivity detector from the GE Tracerlab FX Cpro module; column: Ascentis RP-Amide 250 × 10 mm (Supelco).

16. Isolation of the product using 10 mM AcONa/50 mM AcOH/0,1 g/L ascorbic acid, flow rate 4 mL/min;

17. Manual collection of the product-containing fraction into a collection vial;

18. Transfer the product solution from a collection vial into a sterile, filter-vented final product vial via 0.22 µm sterile membrane filter using a flow of N_2_.

## 4. Conclusions

We demonstrated for the first time that alcohols are suitable co-solvents for Cu-mediated ^18^F-fluorination of iodonium salts. The developed evaporation-free radiofluorination protocol enables a simple, fast and efficient preparation of labeled aromatics and is amenable to automation. The practicality of the protocol was confirmed by the high-yielding production of clinical doses of 2–4-[^18^F]fluorophenylalanines as well as by its direct implementation in a remote-controlled synthesis unit. Additionally, the novel method avoids significant racemization observed if the conventional “minimalist” protocol is applied for the preparation of *ortho*-[^18^F]fluoro-substituted aromatic amino acids.

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
