# Peer review of "Alcohol-Supported Cu-Mediated 18F-Fluorination of Iodonium Salts under “Minimalist” Conditions"

_molecules, 2019, doi:10.3390/molecules24173197_

Round 1
Reviewer 1 Report
The authors in this joint effort from Cologne/Juelich and St.Peterburg successfully demonstrated an improved route for rapid and robust radiosynthesis of several aromatic F-18-fluorinated compounds without the necessity of an evaporation step that is time-consuming and can lead to racemization. The scientific steps and optimization processes are clearly described and easy to follow; the topic is of high relevance and the presentation is accurate. Hence, this is the kind of composition of a manuscript a reviewer likes to read more often.
My only minor concerns are the following:
in respect of ststistical clarity: in Figure 1, part A no error bars are shown although similar data in Fig.2, part A give a visual representation of the deviation of data. Also add the info whether error bars represent standard deviations or CoV or ...? Therefore, I suggest to add a separate paragraph in the experimental section describing all statistic related features very briefly. Figure 3: did the authors also try even shorter reaction times, e.g. 1min? Did the reaction include or exclude time necessary to heat up the solution? page 17, line 457: why did the authors still use 20min reaction time for the manual preparation of the FPhe-derivatives when they showed that 5min lead to the same results? -15min would lead to a considerable gain in overall yield due to reduced decay of F-18... page 18, line 489: in step 13 it says that 12N HCl was used (also on page 10, line 240), but in the caption of figure 6 10N HCl are stated. Please correct this discrepancy. Also, please comment on any problems the appplication of such a high-molarity hydrochloric acid has brought if any, e.g. did the valves in this pathway (V10) break earlier? Were there any concerns with the tubings in this pathway? Was there a replacement of parts necessary sooner than usual?Author Response
Reviewer 1:
in respect of ststistical clarity: in Figure 1, part A no error bars are shown although similar data in Fig.2, part A give a visual representation of the deviation of data.
Reply:As stated in the Manuscript, lines 134, 135: “Experiments with 1·OTs were carried once (except for MeOH, n>20 and nBuOH, n=2) and with 1·BF4in triplicate.” Consequently, in Figure 1A, no error bars are given.
Also add the info whether error bars represent standard deviations or CoV or ...?
Reply:Added to the captions to Figures 1–3: “Error bars represent standard deviations (SDs)”. Added to the caption to Table 1: “For experiments which were carried out at least in triplicate mean values ± standard deviations (MV±SD) are given.”
Therefore, I suggest to add a separate paragraph in the experimental section describing all statistic related features very briefly.
Reply:Added to the manuscript, line 410:”Standard deviations were calculated using standard formulae.”
Figure 3: did the authors also try even shorter reaction times, e.g. 1min?
Reply:No, we did not try reaction times than 5 min because they are difficult to reliably reproduce in an automated synthesis module.
Did the reaction include or exclude time necessary to heat up the solution?
Reply:A preheated aluminum block was used for the optimization experiments. Consequently, the heat up time was neglected.
page 17, line 457: why did the authors still use 20min reaction time for the manual preparation of the FPhe-derivatives when they showed that 5min lead to the same results? -15min would lead to a considerable gain in overall yield due to reduced decay of F-18...
Reply:Indeed, the reduction of the preparation time on 15 min should give a 9% gain of n.d.c. RCYs. Unfortunately, we studied radiolabeling kinetics quite recently, already after the manual syntheses of [18F]FPhes had been carried out.
page 18, line 489: in step 13 it says that 12N HCl was used (also on page 10, line 240), but in the caption of figure 6 10N HCl are stated. Please correct this discrepancy.
Reply:Corrected.
Also, please comment on any problems the appplication of such a high-molarity hydrochloric acid has brought if any, e.g. did the valves in this pathway (V10) break earlier? Were there any concerns with the tubings in this pathway? Was there a replacement of parts necessary sooner than usual?
Reply:We use valves for aggressive media from “Bürkert” (Ingelfingen, Germany), type 0127, order nr.: 79901, with the separating diaphragm from ETFE. They demonstrate very good stability to concentrated HCl also at elevated temperatures. Also Teflon tubes, which are used in our module, seems to be quite stable to HCl.
Reviewer 2 Report
This paper builds on the groups previous work with alcohol enhanced and minimalist 18F-fluorination of iodonium salts, which is in turn based upon methodology developed by the Sanford lab.
The main innovation of the current work is the elimination of the azeotropic dry down of the 18F-fluoride, which should be of interest to PET radiochemists using the methods for their research. Suggest just mnor edits:
Line 47 -- no carrier added should be no-carrier-added
line 68 - 69 -- you should mention Cu-mediated fluorination of iodonium salts generated in situ and cite Sanfords paper (Org Lett 2017, 19, 3939-3942)
line 71 -- "in the majority of cases" is a strong statement that I don't agree with. Suggest changing to "in some cases"
line 72 --refs 11 - 15 -- need to add a couple more citations (Scientific Reports 7, Article number: 233 (2017) and JLCR 61,228-236 (2018))
line 85 -- to to -- need to delete to
Scheme 1 -- method A. As I read the original report, they used 55.5 GBq 18F as proof of concept, which I would consider upscaling. So like the comment above, not sure I agree that upscaling was demonstrated in this work.
Scheme 1 -- method C -- √√ upscaling -- delete one check mark.
Throughout paper -- should OTos be OTs? I think the latter is more common abbreviation.
Lines 136 - 137 -- Please add some discussion on why your methods works with DMF/MeOH, but not DMF/H2O
Line 168 -- "was studied" should be "were studied"
Lines 193 - 197 -- state these results are consistent with prior reports and cite your paper and Sanfords.
line 225 -- extra space between out at
line 251 -- period should be after bracket rather than before.
Author Response
Reviewer 2:
Line 47 -- no carrier added should be no-carrier-added
Reply:Corrected.
line 68 - 69 -- you should mention Cu-mediated fluorination of iodonium salts generated in situ and cite Sanfords paper (Org Lett 2017, 19, 3939-3942)
Reply:Corrected, added to the manuscript:”as well as for (aryl)(mesityl)iodonium salts generated in situfrom electron-rich (hetero)aromatics [10]”
"in the majority of cases" is a strong statement that I don't agree with. Suggest changing to "in some cases"
Reply:Corrected.
refs 11 - 15 -- need to add a couple more citations (Scientific Reports 7, Article number: 233 (2017) and JLCR 61,228-236 (2018))
Reply:The citations have been added to the manuscript.
line 85 -- to to -- need to delete to
Reply:Corrected.
Scheme 1 -- method A. As I read the original report, they used 55.5 GBq 18F as proof of concept, which I would consider upscaling. So like the comment above, not sure I agree that upscaling was demonstrated in this work.
Reply:In the seminal work (ref. 6) Ichiishi et al. isolated 16.5 mCi of the protected intermediate, Piv-6-[18F]FDOPA(OMe)2-OMe, starting from 1.5 Ci [18F]fluoride within 66 min (n.d.c RCY of 1.1%, d. c. RCY of 1.7%), whereas RCCs using small aliquots of 18F–amounted to 31±3% (please, refer to the Scheme 2 and ref. 23 in the original work, as well as to S.19 in the Supporting Information to this paper). We do not think it could be considered as successful upscaling. Similarly, in our hands (ref. 15), application of the original procedure for larger amounts instead of small aliquots of 18F–resulted in a sharp drop of RCCs (from 36–84 to <0.5–5%; please, refer to ref. 15 for the appropriate discussion).
Scheme 1 -- method C -- √√ upscaling -- delete one check mark.”
Reply:Revised.
Throughout paper -- should OTos be OTs? I think the latter is more common abbreviation.
Reply:Corrected.
Lines 136 - 137 -- Please add some discussion on why your methods works with DMF/MeOH, but not DMF/H2O
Reply:Added to the line 137: “presumably, owing to the decomposition of Cu(MeCN)4OTf in the presence of water and/or unfavorably strong solvation of 18F–“.
Line 168 -- "was studied" should be "were studied"
Reply:Corrected.
Lines 193 - 197 -- state these results are consistent with prior reports and cite your paper and Sanfords
Reply:Added to the manuscript: “Accordingly, using the novel protocols, model compounds were prepared in RCCs comparable or better with those obtained applying the original “minimalist” protocol for Cu-mediated radiofluorination [16].” As already stated, the original protocol of Ichiishi et al. works properly only with small aliquots of 18F–(vide supra). Consequently, it is not fully correct to compare the novel method with this procedure here.
line 225 -- extra space between out at
Reply:Corrected.
line 251 -- period should be after bracket rather than before
Reply:Corrected.
Reviewer 3 Report
Neumaier and Krasikova describe an improvement of a known reaction, the Cu-mediated 18F fluorination of iodonium salts.
Line 69: add citation: Taylor, N. J. et al. Derisking the Cu-mediated 18F-fluorination of heterocyclic positron emission tomography radioligands. J. Am. Chem. Soc. 139, 8267-8276 (2017).
The improved method is useful as it does not require evaporation steps which reduce the overal synthesis time. This can indeed be a significant factor to consider in a radiosynthesis and as such it is an improvement to known approaches.
The ability to elute [18F]fluoride with a high efficacy using the reaction solvent, is useful. Neumaier has previously shown a similar optimisation on the copper-mediated 18F-labelling of pinacol arylboronates and boronic acids . In this manuscript they illustrate that a similar approach using alcohols as co-solvents can enhance the elution of [18F]fluoride but also 18F incorporation in the Cu-mediated 18F-fluorination of iodonium salts.
They authors found that using 30% MeOH in DMF as the elution solution was optimal with elution efficacy’s of 90% whilst maintaining high 18F-incorporation (RCC).
The authors have previously described that nBuOH can be used to elute [18F]fluoride (Zischler, J. et al. Chem. Eur. J. 2017, 23, 3251 – 3256 ). Can the authors provide an explanation for why nBuOH is a poor eluent for eluting [18F]fluoride in the reaction investigated in this manuscript?
The authors describe the benefits of using alcoholic solvents in both elution and 18F-incorporation for the Cu-mediated 18F-fluorination of iodonium salts. Other methods using iodonium salts are known which are metal free. Have the authors investigated the effect of alcoholic solvents under metal-free conditions as well, to make [18F]FDOPA for example (A. Maisonial‐Besset et al., Eur. J. Org. Chem., 2018, 7058-7065.).
The authors have not measured the molar activity of the final products. The change of reaction conditions (including the solvent) can have significant effects on the molar activity. The authors must measure the molar activity and compare its value with the original protocol. Most radiolabelled molecules in this manuscript are stable on HPLC and therefore a UV calibration curve can easily be made to measure molar activity.
The authors claim that aerobic conditions were not a prerequisite for 18F-incorporation. This is an advantage for automation.
The authors state: ‘However, surprisingly, no formation of 4-[ 18F]fluoroanisole ([18F]2) was observed in any of the examined solvents including DMA…’ "This somewhat unexpected observation could be possibly explained by the fact that DMF, in contrast to all other studied solvents, is a monodentate O-donor ligand which can stabilize the intermediately formed (Ar)(MesI)Cu(III) 18F complex and facilitate its reductive elimination.’ Have other solvents similar to DMF such as DMPU or DMI been tried? Solvents such as DMI have shown to provide significant improvements to the Cu-mediated 18F-fluorination of aryl boronate esters.
The authors illustrate that very low precursor loadings can be used (3.5 µmol), which is especially useful when the radiolabelling precursor is not trivial to synthesise.
The authors illustrate the applicability of their protocol on the synthesis of 2-4-[18F]FPhes. Unlike previous methods which show significant loss of enantiopurity (Modeman et al. 2-[18F]Fluorophenylalanine Synthesis by Nucleophilic 18F-Fluorination and Preliminary Biological Evaluation. Synthesis, 2019, 51, 664–676), under the conditions described in the manuscript, an enantiomeric excess of >95% is obtained. This improvement alleviates the need for separation the enantiomers on a chiral HPLC column. The ability to access 18F-labelled molecules in an enantiopure fashion is important.
Supporting information: No concerns.
Overall: After the the authors address our concerns (measurement of molar activity, solvent effect etc), this manuscript can be accepted in Molecules.
Author Response
Reviewer 3:
Line 69: add citation: Taylor, N. J. et al. Derisking the Cu-mediated 18F-fluorination of heterocyclic positron emission tomography radioligands. J. Am. Chem. Soc. 139, 8267-8276 (2017).
Reply:Added.
They authors found that using 30% MeOH in DMF as the elution solution was optimal with elution efficacy’s of 90% whilst maintaining high 18F-incorporation (RCC).
The authors have previously described that nBuOH can be used to elute [18F]fluoride (Zischler, J.et al. Chem. Eur. J. 2017, 23, 3251 – 3256 ). Can the authors provide an explanation for why nBuOH is a poor eluent for eluting [18F]fluoride in the reaction investigated in this manuscript?
Reply:Cu-Mediated radiofluorination of boronic substrates is boosted by the addition of alcohol (please, see Fig. 3 in ref. 22). On the contrary, Cu-mediated radiofluorination of iodonium salts tolerates the presence of some alcohol amount in the reaction medium. An increase of alcohol content over 15% of the reaction volume results in a fast decrease of 18F-incorporation. Consequently, in contrast to ref. 22, where salt solutions in purealcohols (400 µL) were used for elution, in this work salt solutions in 20% ROH in DMF(620 µL; 120 µL volume of salt solutions in pure ROH was insufficient for the efficient elution of 18F–) were applied in the optimization study. Furthermore, whereas basic salts like Et4NHCO3and K2CO3usually provide a good elution efficiency, in the case of non-basic salts the recovery rate is dependent on the nature of the salt. For example, 4-(PhI+TfO–)PhCHO was a better eluting agent than 4-(PhI+I–)PhCHO (please, refer to Table S2 in the Supporting Information to ref. 16). It could also be an additional reason, why (4-MeOPh)(Mes)I+OTs–used in this study, provided a lower recovery rate compared to salts used in ref. 22.
The authors describe the benefits of using alcoholic solvents in both elution and 18F-incorporation for the Cu-mediated 18F-fluorination of iodonium salts. Other methods using iodonium salts are known which are metal free. Have the authors investigated the effect of alcoholic solvents under metal-free conditions as well, to make [18F]FDOPA for example (A. Maisonial‐Besset et al., Eur. J. Org. Chem., 2018, 7058-7065.).
Reply:During the development and optimization of the Cu-free “minimalist” protocol (ref. 16) the deleterious effect of residual MeOH on the 18F-incorporation was observed. Nevertheless we tested the applicability of the novel protocol for the Cu-free radiolabeling. Radiofluorination of (Mes)2I+TsO–and (Mes)(4-MeOPh)I+TsO–under “minimalist” conditions afforded [18F]fluoromesitylene in RCCs of 27 and 51%, respectively. By using the novel protocol the 18F-incorporation rate did not exceed 5%.
The authors have not measured the molar activity of the final products. The change of reaction conditions (including the solvent) can have significant effects on the molar activity. The authors must measure the molar activity and compare its value with the original protocol. Most radiolabelled molecules in this manuscript are stable on HPLC and therefore a UV calibration curve can easily be made to measure molar activity.
Reply:We have determined the molar activity for 4-[18F]FPhe. Added to the manuscript, lines 245-246: “The molar activity determined for 4-[18F]FPhe (2.22 GBq), which was produced from the iodonium salt 13, amounted to 207 GBq/µmol.”
Added to the Supporting Information:
Molar activity calculation:The molar activity (GBq/μmol) was calculated by dividing the radioactivity of the 18F-labeled product by the amount of the unlabeled tracer determined from the peak area in a UV-HPLC chromatograms (λ=254 nm). The solution of 4-[18F]FPhe obtained after HPLC purification was concentrated under reduced pressure, the residue was dissolved in a small amount of the HPLC eluent. The resulting solution was completely injected into the HPLC system. The peak area was determined and the amount of 4-FPhe was calculated according to the calibration curve. The molar activity of 4-[18F]FPhe (2.22 GBq) was determined to 207 GBq/μmol.
Calibration curve for 4-[18F]FPhe:
|
C, µmol |
Peak area (mAU) |
|
0.012 |
0.073 |
|
0.008 |
0.051 |
|
0.006 |
0.037 |
|
0.003 |
0.018 |
|
0.001 |
0.004 |
The authors state: ‘However, surprisingly, no formation of 4-[ 18F]fluoroanisole ([18F]2) was observed in any of the examined solvents including DMA…’ "This somewhat unexpected observation could be possibly explained by the fact that DMF, in contrast to all other studied solvents, is a monodentate O-donor ligand which can stabilize the intermediately formed (Ar)(MesI)Cu(III) 18F complex and facilitate its reductive elimination.’ Have other solvents similar to DMF such as DMPU or DMI been tried? Solvents such as DMI have shown to provide significant improvements to the Cu-mediated 18F-fluorination of aryl boronate esters.
Reply:We have additionally tested radiolabeling of 1·OTs in MeOH/DMI and MeOH/DMPU. No formation of [18F]2has been observed in these experiments. Added to the manuscript, line 161: “DMI, DMPU”.
